# Safety and contribution of elderly whole blood donors after raising the upper age limit: Hemovigilance data from a Chinese region from 2012 to 2023

Chengli Yang[1], Xiaoying Jiang[1], Yuzhe Ren[1], Xiaobing Zhu[1]*, Junhong Yang [1,2]*

**1** Chongqing Blood Center, Jiulongpo, Chongqing, China, **2** Working Party on Hemovigilance of the Chinese Society of Blood Transfusion, Jiulongpo, Chongqing, China

* yangjunhong2013@163.com (JY); 32580063@qq.com (XZ)

## Abstract

### Background

In 2012, China raised the upper age restriction for blood donors from 55 to 60 years old. This study analyzed the impact of raising the upper age restriction on whole blood donor health, contribution to blood supply, and safety of blood.

### Methods

The blood collection and donor hemovigilance data of the Chongqing Blood Center from 2012 to 2023 were analyzed to evaluate the safety of elderly blood donors. To evaluate the impact on blood donor and blood component safety, the number of blood donors and donations, the rate and causes of deferrals, donor health (vasovagal reaction risk), and the screening results of transfusion-transmitted infectious diseases of elderly blood donors (56–60 years) were compared to young blood donors (18–55 years).

### Results

During the 12-year period, the proportion of elderly blood donors in whole blood donations increased 10-fold (0.19% to 1.89%), and the proportion of blood collections increased by the same factor (0.20% to 2.15%). Lower deferral rates were observed among older male donors compared to younger ones, whereas the reverse was found in females. A lower incidence of vasovagal reactions and positive HBV, HCV, and syphilis screening rates was found among male and female elderly blood donors compared to young blood donors. In contrast, the positive HIV reaction rate of older female donors was higher than that of young female blood donors.

**Data availability statement:** All relevant data are within the paper and its Supporting Information files.

**Funding:** This study was funded by the Society of Chongqing Blood Transfusion Hualan Biological Research Fund (CQSX-HL202401) and the Chongqing Blood Center Research Fund (2024MPJH07). The funders had no role in study design, data collection and analysis, decision to publish, or preparation of the manuscript.

**Competing interests:** The authors have declared that no competing interests exist.

## Conclusion

This study finds an increasing frequency of donors aged 56–60. Deferral rates are lower among older male donors compared to young, but the reverse is seen in females. In addition, VVR rates and HCV, HBV and syphilis incidences are lower in old donors, but HIV rates appear highest among older donors.

## Introduction

With the aging global population, the proportion of working-age blood donors is gradually decreasing. The proportion of the population over 65 years of age increased from 9.4% in 2012 to 15.4% in 2023 in China [1]. Therefore, the blood supply in China faces increasing pressure due to the decrease in blood donors and the increasing demand. Recently, multiple studies have investigated the safety of elderly blood donors and their donated blood; growing evidence supports blood donation from healthy elderly people [2].

Biomedical Excellence for Safer Transfusion Collaborative (BEST) studied the vasovagal reactions (VVRs) in blood donors over the age of 70 in five developed countries [3]. Speedy et al. analyzed the incidence of VVRs among blood donors after removing upper age restrictions for returning donors and increasing the new donor upper age restriction in Australia [4]. Quee et al. reported that after the upper age limit for blood donation was raised from 69 to 79 years in the Netherlands, they analyzed the willingness of elderly blood donors to raise the age limit for blood donation from the perspective of blood donors, identified their motivations and obstacles for blood donation, and described their experiences and adverse reactions associated with blood donation [5]. The following conclusions are drawn from these studies. Elderly blood donors have a strong willingness to donate blood, with an increasing contribution to the blood supply; in addition, no adverse effects on the health of elderly blood donors have been found due to blood donation. The 2022 Serious Hazards of Transfusion (SHOT) report analyzed adverse reactions to blood donation in donors over the age of 70, revealing higher rates of bruising and delayed bleeding among regular whole blood donors compared to donors aged 25–70 [6]. Notably, the safety of elderly blood donors and the blood they donate needs to be given attention. A recent BEST study analyzed in vitro hemolysis rates of red blood cells donated from two age groups of blood donors (16–19 years vs. ≥ 75 years) from three blood centers in the United States and Canada. Red blood cells from adolescent donors showed a twofold higher rate of oxidative hemolysis compared to red blood cells from elderly donors [7]. Current blood transfusion medicine research aims to understand the relationship between the quality of blood products and the age of the donors, as well as the influence of blood components from blood donors of different ages on patients' health. Hence, the safety of blood components should be explored from the perspective of blood donor [8].

In 2012, China revised its national standard Whole Blood and Component Donor Selection Requirements (GB 18467−2011), raising the age limit for repeat blood

donors from 55 to 60, while the upper age limit for first-time blood donors remained at 55 years old [9]. Following the policy adjustment, no study has investigated the safety of elderly blood donors in China. Therefore, the blood donor hemovigilance data from Chongqing, China, were analyzed to provide a reference for evaluating the effects of the revised age standard for blood donors.

## Materials and methods

### Study population

The service scope of Chongqing Blood Center covers the population in the main city area of Chongqing, which comprises the 10 districts with the highest population density in the city, with a resident population of about 10 million. This area accounts for about 50% of Chongqing's total blood supply. Our previous article described China's blood donor health screening standards in detail. Specifically, the interval between two whole blood donations should be no less than six months [10]. A comparative analysis of the data of elderly blood donors (56−60 years old) and 18−55 years old blood donors was conducted across the 12 years (01/01/2012 to 31/12/2023) since the revision of the upper age limit for blood donors. The 12 years of blood donor safety surveillance data were accessed on 25/10/2024 and were retrospectively analyzed. The results illustrate the contribution of elderly blood donors to the blood supply and provide an analysis of donation safety in this population. This study was approved by the Ethics Committee of Chongqing Blood Center (No.2024-018-01).

### The contribution of elderly donors to the blood supply

In China, donors can choose to donate 200mL, 300mL, or 400mL of whole blood at each donation and the interval between two donations should be no less than 6 months. The trends of elderly blood donors, donations, and the frequency of blood donations of different age groups were analyzed. These findings show the contribution of elderly blood donors to blood supply following the raising of the upper age restriction of blood donors.

### Deferrals of blood donation

The causes of deferrals of blood donation were analyzed by the objective indicators of pre-blood donation detection. The Whole Blood and Component Donor Selection Requirements (GB 18467−2011) stipulate that the hemoglobin (Hb) levels of blood donors must be tested before blood donation, requiring male and female Hb levels of at least 120g/L and 115g/L, respectively. To reflect the actual blood collection process, plasma turbidity and alanine aminotransferase (ALT) levels before blood donation were also measured. After the health consultation for blood donors was completed, a sample was collected to quantitatively and rapidly test the ALT and plasma turbidity of each donor. We utilized QL1000C Automatic Blood Donation Screening Analyzer (Hope Medical) to measure plasma turbidity and ALT levels. The assessment of plasma turbidity was conducted using turbidity analysis, which categorizes the degree of turbidity into three grades— Grade I (mild), Grade II (moderate), and Grade III (severe). Blood donation was permitted for donors with plasma turbidity classified as Grade I or Grade II. ALT levels were determined using the rate method, and donors are eligible to donate blood if their ALT level is ≤ 50 U/L. Therefore, the data of blood donors of different ages who had to postpone blood donation due to low Hb concentration, plasma turbidity, high ALT, or other causes were analyzed.

### Blood donor safety evaluation

Data on blood donation adverse reactions were collected as from 2020 according to the International Society Of Blood Transfusion (ISBT) standards for blood donation adverse reactions [11]. The proportion of blood donors with VVRs in different age groups (18–55 years vs 56–60 years) and gender in 2020–2023 was analyzed to evaluate the safety of elderly blood donors.

## Blood safety evaluation

The results of transfusion-transmitted infectious disease (TTID) screening indicators were used for evaluation. Screening programs for TTID in China include HBV (HBsAg + HBV DNA), HCV (anti-HCV + HCV RNA), HIV (anti-HIV + HIV RNA), and Syphilis antibodies. The blood samples of donors were simultaneously tested by serological enzyme-linked immunosorbent assay (ELISA) and viral nucleic acid test (NAT). NAT testing performed individually by Tigris (Grifols). Regarding the TTID results, we did not conduct a confirmatory test to determine if they were positive; instead, they were unconfirmed reactive results. Blood with reactive ELISA or NAT test results was judged as unqualified. The TTID screening data from 2012 to 2023 were analyzed to assess the safety of blood from elderly donors.

## Statistical analysis

All data was exported from the blood information management system after approval. At the time the authors obtained the data, donor information had been anonymized. Individual participants could not be identified from the obtained information. R software (version 4.2.3) was used for all statistical analyses. GraphPad software (GraphPad Prism 9) was employed to produce statistical graphs. The comparison of rates was conducted using the chi-square test, and a P value less than 0.05 was considered statistically significant.

## Results

### The contribution of elderly donors to the blood supply

A total of 1,435,805 people donated blood in Chongqing Blood Center from 2012 to 2023. Among them, 11,420 blood donors were aged 56–60, accounting for 0.80% of the total blood donors. The proportion of 56–60-year-old blood donors increased from 0.19% in 2012 to 1.89% in 2023 (Table 1). Among elderly blood donors, the proportion of males increased from 0.21% in 2012 to 1.84% in 2023; the proportion of female blood donors increased from 0.16% in 2012 to 1.95% in 2023 and surpassed the proportion of male blood donors in 2023. The number of blood donors aged 56–60 accounted for 0.83% of the total number of male and 0.73% of the total number of female blood donors respectively, and the difference was statistically significant (p < .000001).

A total of 2,447,184 units of whole blood were collected from 2012 to 2023. Among them, 22,320.5 units were from donors aged 56–60, accounting for 0.91% of the total collections. Blood collected from elderly donors accounted for a growing proportion year by year, increasing from 0.20% in 2012 to 2.15% in 2023. The proportion of blood collected from males increased from 0.22% in 2012 to 2.00% in 2023. For women, the proportion also increased from 0.17% to 2.38% over the same period. Notably, a larger number of elderly women donated blood compared to elderly men in 2023 (Fig 1). Among male blood donors, the proportion of those aged 56–60 who donated 400 ml of whole blood was 83.72%, significantly higher than 70.10% for those aged 18–55, p < 0.000001. Similarly, among female blood donors, the proportion of those aged 56–60 who donated 400 ml was 76.99%, significantly higher than 47.86% for those aged 18–55, p < 0.000001 (S1 Table).

The trend of the proportion of whole blood donors aged 56–60 from 2012 to 2023 is shown in Fig 2. The distribution trends for men (Fig 2A) and women (Fig 2B) were similar. As age increased, the proportion of donors gradually decreased. Donors aged 56 and 57 accounted for the largest proportion, making up 60% to 80% of the donors in the 56–60 age group. Donors aged 58 accounted for 8.94% to 23.67%, and those aged 59 accounted for 6.29% to 21.77%.

### The deferral rate of elderly donors

The deferral rates for 56–60-year-old donors and 18–55-year-old donors were both 1.43%. As displayed in Fig 3, the deferral rate was lower for older male donors than for younger donors (1.50% vs 1.76%, p = 0.008534), while the reverse was true for female donors (1.34% vs 1.07%, p = 0.001646). Among male donors (Fig 3A), the deferral rate due to low

**Table 1. Number of donors, number of donations, and average number of donations per donor stratified by age and sex, 2012–2023.**

| Year | | Number of Donors | | | Number of Donations | | | Average Number of Donations per Donor | |
|---|---|---|---|---|---|---|---|---|---|
| | | Total | 18-55 | 56-60 | Total | 18-55 | 56-60 | 18-55 | 56-60 |
| 2012 | All donors | 98,837 | 98,645 | 192 | 102,744 | 102,551 | 193 | 1.04 | 1.01 |
| | Males | 54,895 | 54,774 | 121 | 57,041 | 56,919 | 122 | 1.04 | 1.01 |
| | Females | 43,942 | 43,871 | 71 | 45,703 | 45,632 | 71 | 1.04 | 1.00 |
| 2013 | All donors | 92,993 | 92,633 | 360 | 97,758 | 97,357 | 401 | 1.05 | 1.11 |
| | Males | 49,545 | 49,320 | 225 | 52,133 | 51,882 | 251 | 1.05 | 1.12 |
| | Females | 43,448 | 43,313 | 135 | 45,625 | 45,475 | 150 | 1.05 | 1.11 |
| 2014 | All donors | 91,786 | 91,352 | 434 | 95,333 | 94,855 | 478 | 1.04 | 1.10 |
| | Males | 49,865 | 49,597 | 268 | 51,845 | 51,548 | 297 | 1.04 | 1.11 |
| | Females | 41,921 | 41,755 | 166 | 43,488 | 43,307 | 181 | 1.04 | 1.09 |
| 2015 | All donors | 102,630 | 102,142 | 488 | 107,343 | 106,799 | 544 | 1.05 | 1.11 |
| | Males | 54,591 | 54,303 | 288 | 57,247 | 56,925 | 322 | 1.05 | 1.12 |
| | Females | 48,039 | 47,839 | 200 | 50,096 | 49,874 | 222 | 1.04 | 1.11 |
| 2016 | All donors | 115,440 | 114,812 | 628 | 122,875 | 122,178 | 697 | 1.06 | 1.11 |
| | Males | 58,421 | 58,067 | 354 | 62,495 | 62,104 | 391 | 1.07 | 1.10 |
| | Females | 57,019 | 56,745 | 274 | 60,380 | 60,074 | 306 | 1.06 | 1.12 |
| 2017 | All donors | 121,732 | 121,174 | 558 | 130,165 | 129,529 | 636 | 1.07 | 1.14 |
| | Males | 61,807 | 61,497 | 310 | 66,282 | 65,925 | 357 | 1.07 | 1.15 |
| | Females | 59,925 | 59,677 | 248 | 63,883 | 63,604 | 279 | 1.07 | 1.13 |
| 2018 | All donors | 119,647 | 119,080 | 567 | 128,012 | 127,385 | 627 | 1.07 | 1.11 |
| | Males | 60,551 | 60,231 | 320 | 65,031 | 64,677 | 354 | 1.07 | 1.11 |
| | Females | 59,096 | 58,849 | 247 | 62,981 | 62,708 | 273 | 1.07 | 1.11 |
| 2019 | All donors | 130,309 | 129,497 | 812 | 139,193 | 138,309 | 884 | 1.07 | 1.09 |
| | Males | 67,474 | 67,035 | 439 | 72,271 | 71,794 | 477 | 1.07 | 1.09 |
| | Females | 62,835 | 62,462 | 373 | 66,922 | 66,515 | 407 | 1.06 | 1.09 |
| 2020 | All donors | 92,993 | 92,633 | 360 | 97,758 | 97,357 | 401 | 1.05 | 1.11 |
| | Males | 49,545 | 49,320 | 225 | 52,133 | 51,882 | 251 | 1.05 | 1.12 |
| | Females | 43,448 | 43,313 | 135 | 45,625 | 45,475 | 150 | 1.05 | 1.11 |
| 2021 | All donors | 136,277 | 134,616 | 1,661 | 144,387 | 142,610 | 1,777 | 1.06 | 1.07 |
| | Males | 70,491 | 69,577 | 914 | 75,034 | 74,048 | 986 | 1.06 | 1.08 |
| | Females | 65,786 | 65,039 | 747 | 69,353 | 68,562 | 791 | 1.05 | 1.06 |
| 2022 | All donors | 125,407 | 123,375 | 2,032 | 130,987 | 128,838 | 2,149 | 1.04 | 1.06 |
| | Males | 64,771 | 63,660 | 1,111 | 67,976 | 66,788 | 1,188 | 1.05 | 1.07 |
| | Females | 60,636 | 59,715 | 921 | 63,011 | 62,050 | 961 | 1.04 | 1.04 |
| 2023 | All donors | 131,831 | 129,365 | 2,466 | 139,250 | 136,617 | 2,633 | 1.06 | 1.07 |
| | Males | 73,718 | 72,388 | 1,330 | 78,110 | 76,672 | 1,438 | 1.06 | 1.08 |
| | Females | 58,113 | 56,977 | 1,136 | 61,140 | 59,945 | 1,195 | 1.05 | 1.05 |
| Total | All donors | 1,359,882 | 1,349,324 | 10,558 | 1,435,805 | 1,424,385 | 11,420 | 1.05 | 1.09 |
| | Males | 715,674 | 709,769 | 5,905 | 757,598 | 751,164 | 6,434 | 1.06 | 1.10 |
| | Females | 644,208 | 639,555 | 4,653 | 678,207 | 673,221 | 4,986 | 1.05 | 1.08 |

hemoglobin levels was significantly higher than that of the young group (0.65% vs 0.15%, p<0.00001). The deferral rate due to plasma turbidity in the elderly was also significantly higher than that of young blood donors (1.89% vs 1.28%, p=0.000377). However, the deferral rate due to excessive ALT in the elderly group was 3.43%, which was significantly

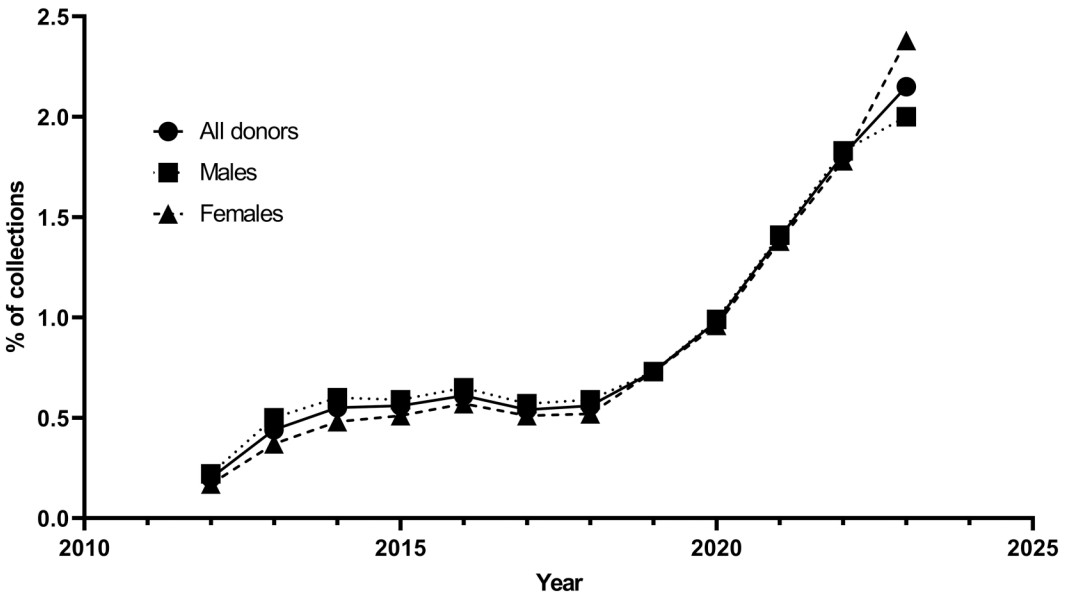

**Fig 1. Trends in blood supply in 56–60-year-old donors, 2012–2023.**

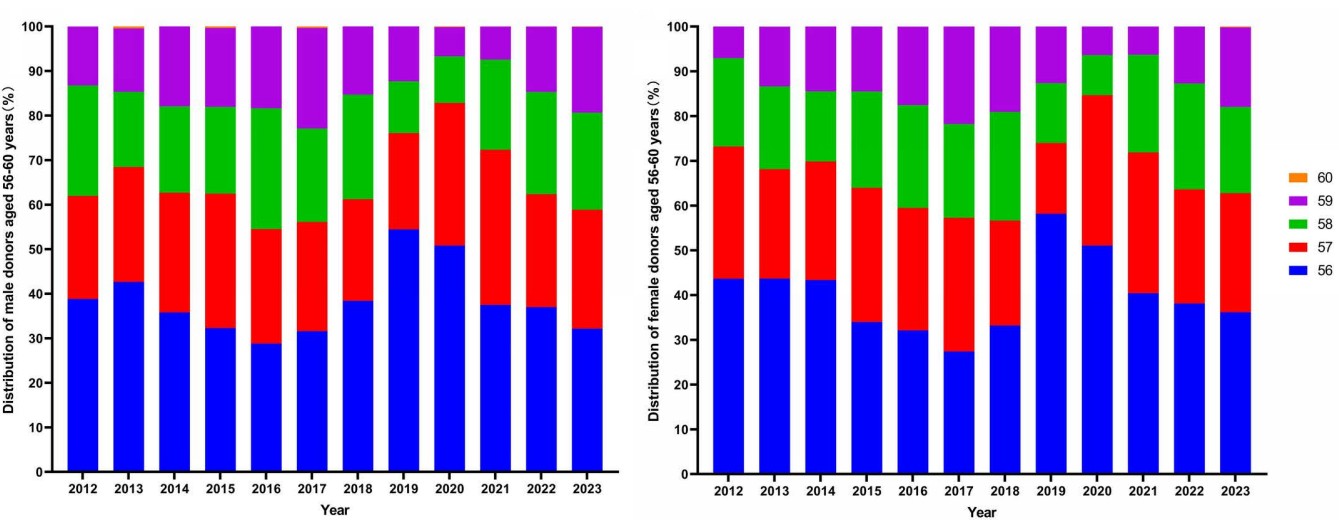

**Fig 2. Trend of population distribution and proportion of 56-60-year-old whole blood donors from 2012 to 2023.**

lower than that in the young group (5.16%), p < 0.000001. Among female donors (Fig 3B), the deferral rate due to low hemoglobin levels in the elderly was 1.94%, which was slightly lower than that in the young group (2.25%), p = 0.217508. The deferral rate due to plasma turbidity in the elderly was also significantly higher than that in young donors (1.06% vs 0.42%, p < 0.000001). The deferral rate of ALT in the older group was 2.34%, which was significantly higher than that in the younger group (1.19%), p < 0.000001. For both male and female donors, the deferral rate due to other reasons was significantly lower in the older group compared to the younger group (0.02% vs 0.46%, p = 0.000022; 0.03% vs 0.41%, p = 0.000440).

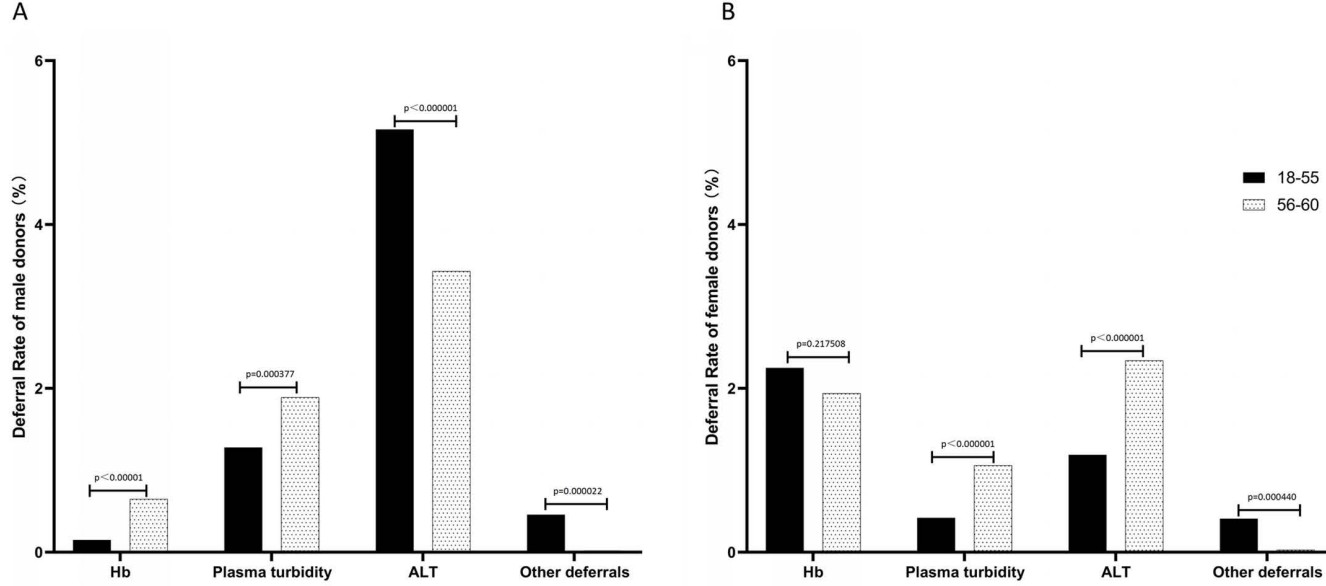

**Fig 3. Deferral rate among whole blood donors.**

## Blood donor safety assessment

The incidence and risk assessment of VVRs in elderly and young blood donors are shown in Fig 4. The incidence of VVRs was 0.16% (6/3863) in elderly male blood donors, which was significantly lower than 1.39% (3741/269390) in young blood donors (p<0.000001). Similarly, the incidence of VVRs in older female donors was 0.39% (12/3097), which was significantly lower than 1.35% (3181/236032) in younger donors (p=0.000005). From the annual data, the incidence of VVRs among 56–60-year-old blood donors was significantly lower than that of 18–55-year-old blood donors from 2021 to 2023.

## Evaluation of blood safety in elderly donors

The results of blood screening for TTID in elderly and young donors are shown in Fig 5. Among male blood donors (Fig 5A), the reactivity rates for HBV, HCV, and syphilis among donors aged 56–60 were significantly lower compared to those aged 18–55 (0.50% vs 1.12%,p=0.000002; 0.06% vs 0.30%, p=0.000502; 0.23% vs 0.51%, p=0.001665). In contrast, the screening reactivity rate for HIV was higher in the elderly compared to young donors (0.36% vs 0.30%, p=0.409430). Similarly, among female blood donors (Fig 5B), the screening reactivity rates for HBV, HCV, and syphilis were significantly lower in 56–60 year old donors than in 18–55 year old donors (0.26% vs 0.85%,p=0.000006; 0.12% vs 0.36%, p=0.005374; and 0.14% vs 0.53%, p=0.000138). Nonetheless, the HIV screening reactivity rate was significantly higher in elderly blood donors (0.54% vs 0.25%, p=0.000065).

## Discussion

The World Health Organization's blood donor selection guidelines state that the upper age limit for blood donors is 65 years [12]. The U.S. FDA did not set an upper age limit for blood donors [3]. Compared with other countries, China has adopted a relatively conservative policy concerning the age limit of blood donors. China's Blood Donation Law aims to ensure the safety of blood donors, which advocates voluntary blood donation by healthy citizens between 18 and 55 years old [13]. In this study, the effects of adjusting donor age to 60 on blood donation health and blood supply were analyzed. The safety of elderly blood donors and the safety of the donated blood were evaluated by analyzing the hemovigilance

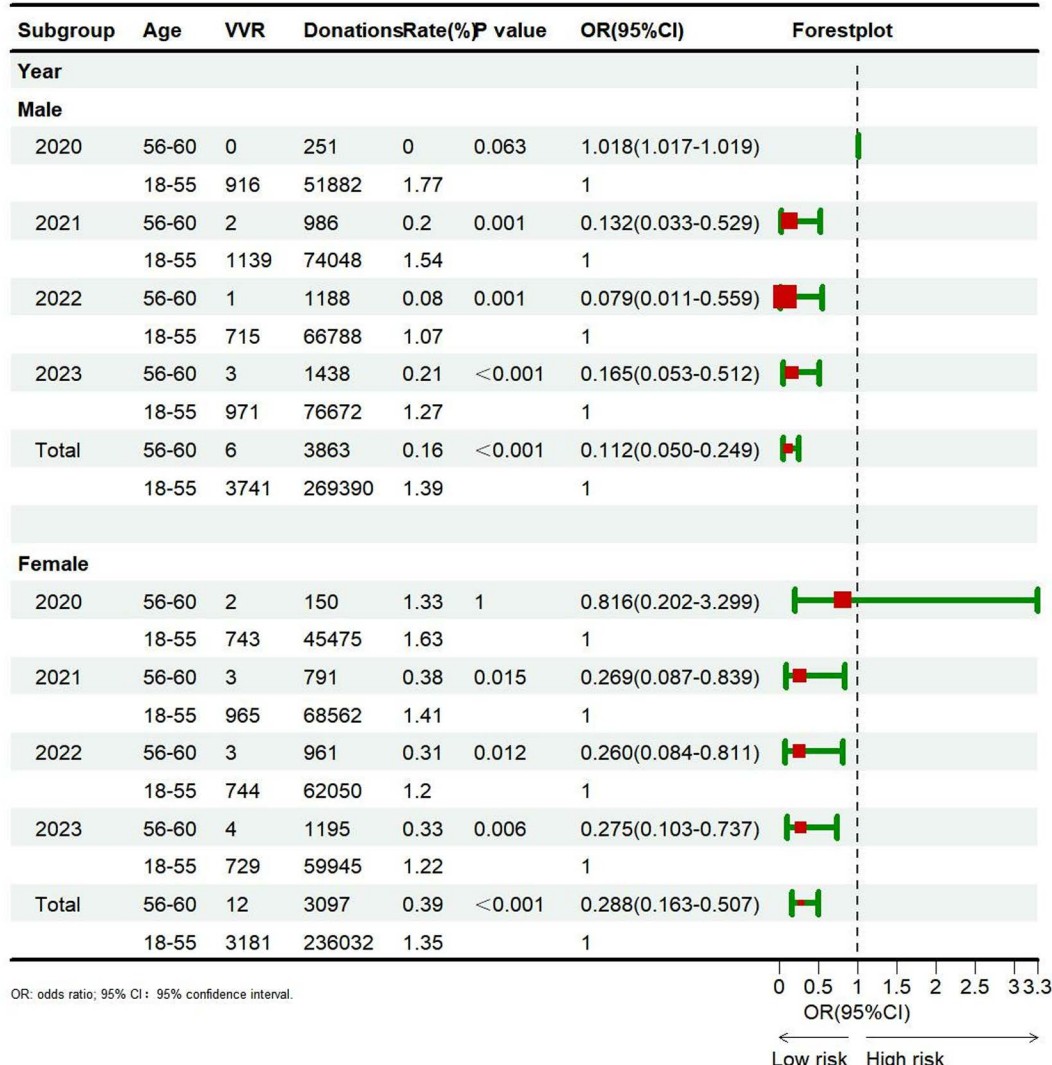

| Subgroup | Age | VVR | Donations | Rate(%) | P value | OR(95%CI) | Forestplot |
|---|---|---|---|---|---|---|---|
| **Year** | | | | | | | |
| **Male** | | | | | | | |
| 2020 | 56-60 | 0 | 251 | 0 | 0.063 | 1.018(1.017-1.019) | |
| | 18-55 | 916 | 51882 | 1.77 | | 1 | |
| 2021 | 56-60 | 2 | 986 | 0.2 | 0.001 | 0.132(0.033-0.529) | |
| | 18-55 | 1139 | 74048 | 1.54 | | 1 | |
| 2022 | 56-60 | 1 | 1188 | 0.08 | 0.001 | 0.079(0.011-0.559) | |
| | 18-55 | 715 | 66788 | 1.07 | | 1 | |
| 2023 | 56-60 | 3 | 1438 | 0.21 | <0.001 | 0.165(0.053-0.512) | |
| | 18-55 | 971 | 76672 | 1.27 | | 1 | |
| Total | 56-60 | 6 | 3863 | 0.16 | <0.001 | 0.112(0.050-0.249) | |
| | 18-55 | 3741 | 269390 | 1.39 | | 1 | |
| | | | | | | | |
| **Female** | | | | | | | |
| 2020 | 56-60 | 2 | 150 | 1.33 | 1 | 0.816(0.202-3.299) | |
| | 18-55 | 743 | 45475 | 1.63 | | 1 | |
| 2021 | 56-60 | 3 | 791 | 0.38 | 0.015 | 0.269(0.087-0.839) | |
| | 18-55 | 965 | 68562 | 1.41 | | 1 | |
| 2022 | 56-60 | 3 | 961 | 0.31 | 0.012 | 0.260(0.084-0.811) | |
| | 18-55 | 744 | 62050 | 1.2 | | 1 | |
| 2023 | 56-60 | 4 | 1195 | 0.33 | 0.006 | 0.275(0.103-0.737) | |
| | 18-55 | 729 | 59945 | 1.22 | | 1 | |
| Total | 56-60 | 12 | 3097 | 0.39 | <0.001 | 0.288(0.163-0.507) | |
| | 18-55 | 3181 | 236032 | 1.35 | | 1 | |

OR: odds ratio; 95% CI: 95% confidence interval.

0　0.5　1　1.5　2　2.5　3　3.3
OR(95%CI)
← Low risk　High risk →

**Fig 4. Incidence and risk assessment of VVRs in elderly and young blood donors.**

data of more than 1.4 million blood donors from the Chongqing Blood Center in the past 12 years. Lower deferral rates were observed among older male donors compared to younger ones, whereas the reverse was found in females. A lower incidence of vasovagal reactions and positive HBV, HCV, and syphilis screening rates was found among male and female elderly blood donors compared to young blood donors. In contrast, the positive HIV reactivity rate of older female donors was higher than that of young female blood donors.

## The contribution of elderly donors to the blood supply

Over the 12 years of data, the proportion of blood donations from elderly donors increased 10-fold, and the proportion of blood collected increased by almost the same amount. A similar increase in proportion was observed in elderly blood donors aged 71 and over in Australia (2.18%); however, the proportion of blood collected was lower than 3.15% [3]. This difference may be attributed to the frequency of blood donation. The average frequency of blood donation among elderly

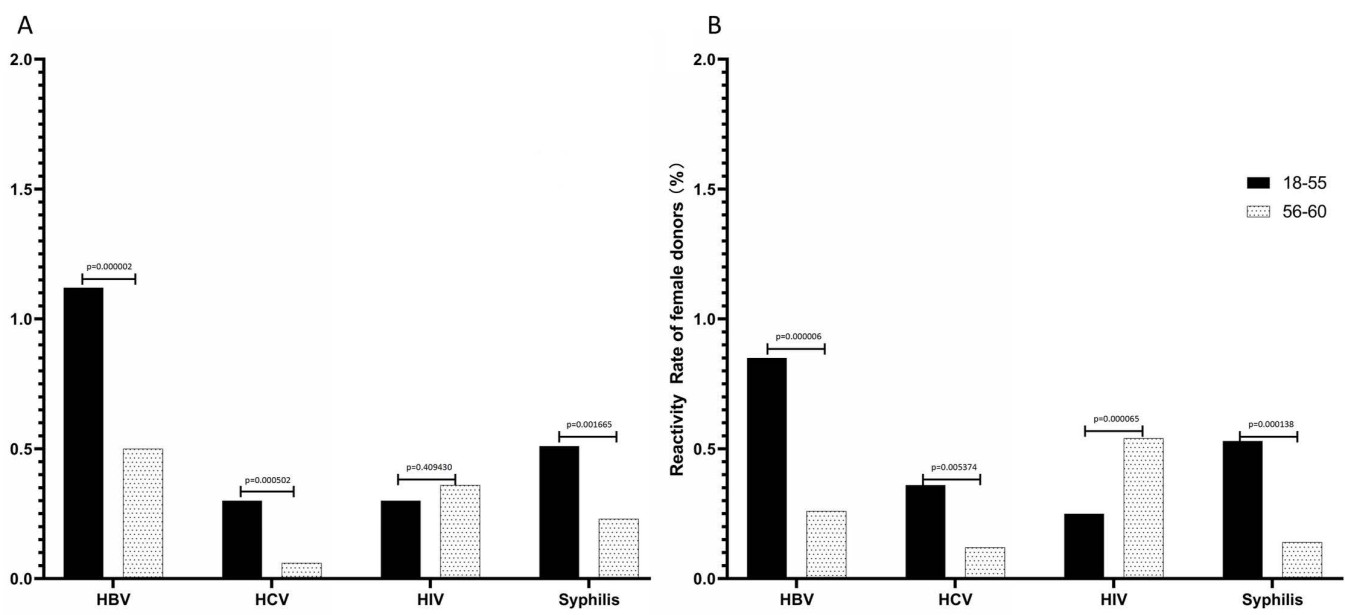

**Fig 5. Results of screening for TTID.**

donors in China is 1.09 donations/donor per annum, while the frequency of blood donation among donors over 71 years old in Australia is 2.55 Donations/donor per annum. This is due to allowable donation frequency though and the Australian rate was probably including plasma/platelet donation which increases the frequency. Moreover, the proportion of female elderly blood donors increased faster than that of male elderly blood donors. After 2013, the elderly have been found to donate whole blood more frequently than young donors, particularly among male donors. In addition, we have not specifically informed every blood donor who is 55 years old or above that they can continue to donate blood until they are 60. It was only when these blood donors came for consultation that they learned they could continue donating blood. Therefore, the proportion of our elderly blood donors is very small.

In terms of age distribution, donors aged 18–55 still account for the majority of blood donors in China, while those over 56 years old only constitute a small proportion, with the number of donors over 58 being even smaller. Since 2020, the proportion of 56-year-old blood donors has shown a slight downward trend, which was consistent for both male and female donors. This downward trend may be attributed to the disruption of blood donation due to COVID-19. However, the number of elderly donors has increased each year, representing a growing proportion of the blood supply.

### The causes of deferrals in male and female elderly donors

In terms of overall deferral rates, no significant difference was observed between older and younger blood donors. However, the rate and causes of deferred blood donation showed significant differences between male and female donors of various ages. This is one of the main findings of our study. Among male blood donors, the overall blood donation deferral rate was lower in elderly blood donors compared to young male blood donors. Conversely, female donors showed the reverse trend. The discrepancy in the results published by BEST [3] may be attributed to different pre-blood donation testing programs. Notably, ALT levels are screened before blood donation, representing a major cause of deferral in male donors, accounting for 73% of all male deferrals. Serum ALT level is related to gender and age, and serum ALT levels are generally higher in the male population than in the female population, reaching a peak at 40−55 years old, and decreasing after 55 years old [14]. The deferral rate caused by low hemoglobin levels was significantly higher in elderly men than in

young men, while no statistical difference was found between elderly women and young women. This is consistent with BEST's findings, which may be related to the decline in normal Hb levels with age. In addition, older men donate blood more frequently, tend to have more chronic health problems, and are also at a higher risk of iron deficiency compared to younger men [15,16]. Another cause of deferred blood donation is plasma turbidity. According to China's national standard "Whole Blood and Component Blood Quality Requirements" (GB 18469−2012), blood with high plasma turbidity cannot be used for clinical infusion [17]. In transfusion medicine, lipid-rich donations pose two major challenges: interference from turbid plasma in laboratory testing and concerns about the safety of extracting blood components from hypertriglyceridemic blood [18]. Optical quantitative detection can effectively reduce blood scrapping due to the collection of plasma with high lipid levels, thereby reducing resource waste [19]. Our study indicated that the deferral rate due to plasma turbidity was higher in the elderly than in the younger population in both male and female blood donors; notably, men exhibited a higher deferral rate than women. Compared to other donors, blood donors with high lipid levels typically have poorer cardiovascular status and dietary habits, high triglyceride (TG) levels, and smoking, which are independent risk factors for plasma cloudiness [20]. Age-induced changes in human TG metabolism include increased plasma TG levels and decreased plasma TG clearance after meals, which may lead to age-related plasma turbidity [21].

### The incidence of VVRs is lower in elderly donors than in young donors

The eligibility of elderly people to donate blood has long been debated. These concerns arise from the increased risk of cardiovascular disease with age, as well as adverse reactions associated with the physiological changes of aging, particularly the increased risk of vasovagal reactions [22]. In order to assess the impact of blood donation on the health of elderly blood donors, a comprehensive donor hemovigilance system was established [10]. The safety of elderly blood donors was evaluated by analyzing the incidence of VVRs in elderly and young donors. Data on adverse reactions from both male and female blood donors indicated that, except for the small number of elderly donors in 2020, the risk of VVRs in donors aged 56–60 was lower than that in donors aged 18–55 [OR for men = 0.112 (0.079–0.165), OR for women = 0.288 (0.260–0.275)]. The results are consistent with those of BEST, the Netherlands, Germany, the United States, and Italy [3,5,23–25]. Our study indicates a high safety profile for blood donation from the elderly in China. Donating whole blood volume can affect the incidence of VVR. Our data analysis shows that blood donors aged 56–60 (both male and female) have a higher proportion of donating 400 ml of whole blood, but the incidence of VVR is significantly lower than that of blood donors aged 18–55. Blood donors aged 56–60 are repeat donors and have not developed any serious adverse reactions during previous blood donations. Therefore, repeat donors are familiar with the blood donation process, which can reduce their fear of donating blood and thereby decrease the occurrence of VVRs.

### Blood safety evaluation

Finally, the safety of blood donated from elderly donors was evaluated by analyzing the positive screening rates for transfusion-transmitted diseases. The results revealed that the screening reactivity rates of HBV, HCV and syphilis in the blood of elderly blood donors are significantly lower than those of young blood donors, which is consistent with the results reported by Chang et al. [26]. However, in contrast to previous studies, the HIV screening results of both male and female elderly blood donors were significantly higher than those of young donors. Shi et al. reported that the HIV incidence rate was estimated to be 37.93 per 100,000 person-years (95% CI 30.62–46.97) among first-time donors and 20.55 per 100,000 person-years (95% CI 16.95–24.91) among repeat donors in China. Retired blood donors were an independent risk factor for HIV infection [27]. In recent years, HIV infection has been detected in increasingly older patients, which may also explain the high response rate of HIV screening among elderly blood donors [28].

## Highlights and limitations

Our study has several advantages. First, this is the first study on blood donors and blood safety in China after the adjustment of the upper age limit for blood donors. Our study includes data from over 12 years from more than 1.4 million blood donors. Second, a comprehensive assessment of blood donation deferral was conducted, which was attributed to quantitative testing before donation. Third, the study analyzed the elderly blood donors' contribution to blood supply, blood donor safety, and blood safety, representing a comprehensive study evaluating the safety of elderly blood donors. Nevertheless, the limitations of the present study should be acknowledged. On the one hand, some of our data have only been collected since 2020, such as the data on adverse reactions to blood donation. This is due to China only establishing a blood donor safety monitoring system in 2019. On the other hand, this study is limited to data from a single center due to significant differences in blood donation screening programs, adverse reaction monitoring methods, and other aspects among different blood banks in China; hence, a comprehensive analysis incorporating data from multiple centers was unfeasible. Notably, some indicators will be selected for future multi-center studies to more comprehensively assess the safety of elderly blood donors in China. Finally, following the guidelines to extend the age restriction for blood donation, the blood donors aged 56–60 were all repeat donors. Such demographic characteristics have a certain impact on the outcome, which cannot be resolved within the current regulatory framework.

## Conclusions

This study finds an increasing frequency of donors aged 56–60. Deferral rates are lower among older male donors compared to young, but the reverse is seen in females. In addition, VVR rates and HCV, HBV and syphilis incidences are lower in old donors, but HIV rates appear highest among older donors.

## Supporting information

**S1 Table.**
(XLSX)

## Acknowledgments

The authors thank all those involved in blood collection and blood testing laboratories, as well as everyone who has provided feedback.

## Author contributions

**Conceptualization:** Xiaobing Zhu.

**Data curation:** Xiaoying Jiang, Yuzhe Ren, Xiaobing Zhu.

**Formal analysis:** Xiaobing Zhu.

**Methodology:** Xiaobing Zhu.

**Project administration:** Xiaobing Zhu.

**Writing – original draft:** Chengli Yang.

**Writing – review & editing:** Xiaobing Zhu, Junhong Yang.

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
