## [Decision Letter · Decision Letter 0]

22 Apr 2025

Dear Dr. Yang,

We look forward to receiving your revised manuscript.

Kind regards,

Rena Hirani

Academic Editor

PLOS ONE

Journal Requirements:

2. Thank you for stating the following financial disclosure: [This study was funded by the Society of Chongqing Blood Transfusion Hualan Biological Research Fund (CQSX-HL202401) and the Chongqing Blood Center Research Fund (2024MPJH07).]. 

Reviewers' comments:

Reviewer's Responses to Questions

**Comments to the Author**

1. Is the manuscript technically sound, and do the data support the conclusions?

Reviewer #1: Yes

Reviewer #2: Yes

2. Has the statistical analysis been performed appropriately and rigorously?

Reviewer #1: I Don't Know

Reviewer #2: Yes

3. Have the authors made all data underlying the findings in their manuscript fully available?

Reviewer #1: No

Reviewer #2: Yes

4. Is the manuscript presented in an intelligible fashion and written in standard English?

Reviewer #1: Yes

Reviewer #2: No

Reviewer #1: This is an interesting study investigating the effect of allowing Chinese donors aged 55-60 to donate. As the authors state it is of great importance to conduct such studies to ensure donor health and monitor the effects of changes in donor eligibility criteria.

While I like the study, there are some things that I find are missing before it is ready for publication:

1) I lack a description of the study population in table form including average number donations, age, gender distribution ect.

2) Are the older donors in this study new or experienced donors? And how does this differ from the younger group they are compared to? This should be accounted for in the analyses and in the interpretation of the results in the discussion.

3) A clear description of the statistical methods is missing including how data was treat-ed, models used and software.

4) For the VVR study it is not listed which covariates were included in the logistic(?) re-gression model. Number of donations or some measure of donation experience should be included to account for the healthy donor effect.

5) The healthy donor effect and how this may impact results is not discussed or mentioned which I find strange.

Introduction

Line 53-61: I lack conclusions on the findings from these manuscripts and why they have been mentioned. Also reg. ref. 4+5 please change to last name of the first author instead of first name (Joanna and Franke).

Methods

Line 89-93: Should be in a separate section covering all methods sections.

Line 95-98: What is donation frequency limits in China? Please include this.

Line 113: Please include which assays are used, company etc..

Results

Table 1 should be cohort descriptives and the current table 1 would be easier to comprehend as a figure.

Reviewer #2: Population aging has two impacts on the blood supply. On the one hand, fewer people are eligible to donate blood. On the other hand, the demand for clinical blood use has increased. In recent years, the feasibility of expanding elderly blood donors has been evaluated all over the world.

The problem is particularly prominent in China, a country that accounts for about 20% of the world's population, where the blood donation rate per thousand people was only 12.2‰ in 2024. Faced with the continued pressure on blood supply, the Chinese government relaxed the age limit for blood donors in 2012, expanding the age limit from 55 to 60 years in an effort to improve the blood availability.

However, blood is supplied to clinical patients, so its safety is of vital importance. This safety is reflected in two aspects: first, the safety of blood donors - as an elderly group, whether donating blood will bring them health risks; second, safety of blood products - whether the blood from the elderly can meet the quality requirements of transfusion medicine.

This paper is a retrospective analysis based on these two concerns. In general, the topic is meaningful, the logical structure of the writing is clear, and the conclusions are presented in an appropriate fashion and supported by the data. As an attempt of China's solution, it provides effective data support for solving China's blood supply problem, and also provides important reference for global countries.

However, the paper needs to be improved in the following aspects.

Major revision

1. Written in standard English

Many expressions in this article do not meet the requirements of standard English. It is recommended to seek a native speaker for revision. The expressions that may cause ambiguity, unclear meaning, or sentences that are too long and need to be re-sentenced include but are not limited to:

Title, L26-30, L34-35, L54-61, L64-65, L69: ecology, L70: opposing, L74-75, L88-89, L91-92, L98: increase, L102: content, L186-190, L193: promote, L201: considers, L227-L234, L239: partly, L240, L244: lipid donation, L246: hypertriglyceridemic, L244-246, L251: diet, L252: TG, L257: suitability, L281&282: strengths.

2. Terminology for transfusion medicine

Many of the terms related to transfusion medicine in this article should be standardized, such as “donation” vs donor”, “donation” vs “donation times”. The mixed use of these expressions is particularly reflected in Table 1, L208, L212-218, etc.

Another example is the inconsistency between the "TTDI" in the main text and the title of Table 5. It is recommended to unify it as TTI (transfusion-transmitted infection).

3. A statistical description should be added to the Materials and Methods section.

There are many statistical inference conclusions in the results section of the article, and the methodological information of these statistical analyses should be given in the materials and methods section.

In addition, “significantly” should be used with caution, and it needs to be supported by statistical inferences, such as in the "The deferral rate of elderly donors" section.

And, does the word “lower” in L167 mean statistically lower? Such expressions should be standardized.

Minor revision

1. Regarding charts: It is recommended that at least Figure 2 be in color.

2. Industry standard should be given a standard number, such as “GB 18469-2012”.

3. The age-related changes in triglyceride levels described in [23] do not appear to be sufficient to threaten the requirement for blood donation. It is necessary to carefully examine whether appropriate references have been cited. Other literatures should also be identified.

4. Others

L132-133: “percent” should be “%”, and L138: “II” should be “2”.

L273: “Le Chang et al” should be “Chang et al”, L275: “Ling Shiet al” should be “Shi et al”. L293: “stations” should be “banks”. L343, L370 should be the “National Health Commission of the People's Republic of China” or the “Former Ministry of Health of the People's Republic of China”.

**Do you want your identity to be public for this peer review?** For information about this choice, including consent withdrawal, please see our Privacy Policy

Reviewer #1: No

Reviewer #2: No

---

## [Author Response · Author response to Decision Letter 1]

11 Jun 2025

Evaluation of the safety and contribution of elderly whole blood donors after raising the upper age restriction: a study of donor hemovigilance data in a region of China from 2012 to 2023

PONE-D-25-08734

Dear editor and reviewers:

We would like to thank the editor and all reviewers for your time and efforts in reviewing our manuscript entitled “Evaluation of the safety and contribution of elderly whole blood donors after raising the upper age restriction: a study of donor hemovigilance data in a region of China from 2012 to 2023”. Your comments and suggestions are valuable and helpful for revising and improving our paper, as well as important guidance for our studies in the future. Here, we have revised and re-submit the manuscript for your consideration. We have responded to each point brought up in your comments, and the changes are highlighted in the revised manuscript as well.

Reviewer #1: 

This is an interesting study investigating the effect of allowing Chinese donors aged 55-60 to donate. As the authors state it is of great importance to conduct such studies to ensure donor health and monitor the effects of changes in donor eligibility criteria.

While I like the study, there are some things that I find are missing before it is ready for publication:

1.I lack a description of the study population in table form including average number donations, age, gender distribution ect.

Response:

Thank you for your attention to it. In the revised manuscript, we modified the content of Table 1 to feature information such as the age, gender and average blood donation volume of blood donors from 2012 to 2023. The contribution of elderly blood donors to the blood supply is visualized in Figure 1.

2.Are the older donors in this study new or experienced donors? And how does this differ from the younger group they are compared to? This should be accounted for in the analyses and in the interpretation of the results in the discussion.

Response:

Thank you for your nice suggestion. We added this part in the limitations of the discussion section of the revised manuscript. The content is as follows: In China, blood donors aged 56 to 60 are all experienced donors, because only those with blood donation experience can participate in blood donation after the age of 55. Therefore, such demographic characteristics will have a certain impact on the outcome, but it cannot be resolved within the current regulatory framework.

3.A clear description of the statistical methods is missing including how data was treated, models used and software.

Response:

Thank you for your advice. In the Materials and Methods section, we have added a description of the statistical methods: All data is exported from the blood information management system after approval. At the time the authors obtained the data, donor information had been anonymized. The authors obtained information during or after data collection that could not identify individual participants. R software (version 4.2.3) was used for all statistical analyses..GraphPad software (GraphPad Prism 9) was used to produce statistical graphs. The comparison of rates was conducted using the chi-square test, and a P value less than 0.05 was considered statistically significant.

4.For the VVR study it is not listed which covariates were included in the logistic(?) re-gression model. Number of donations or some measure of donation experience should be included to account for the healthy donor effect.

Response:

Thank you for your suggestion. Since this article mainly explores the impact of age on the safety of blood donors, we only compared the incidence of VVRs among blood donors of different age groups in the text to assess the safety of elderly blood donors, and did not conduct a regression analysis. As per your suggestion, we have supplemented the total number of blood donations and the specific data of VVRs in the results section.

5.The healthy donor effect and how this may impact results is not discussed or mentioned which I find strange.

Response:

Thank you for your reminder. We added the content of the healthy blood donor effect in the discussion section of VVRs. Blood donors aged 56-60 are experienced donors, and the donors themselves have declared that they have not experienced any serious adverse reactions during previous blood donation procedures. This indicates that these donors have trust and confidence in the blood donation process, which also has a certain impact on the results.

6.Others

(1)Introduction

Line 53-61: I lack conclusions on the findings from these manuscripts and why they have been mentioned. Also reg. ref. 4+5 please change to last name of the first author instead of first name (Joanna and Franke).

Response:

Thank you for your reminder. In this presentation, the mention of these studies is intended to illustrate that the health of elderly blood donors is receiving increasing attention. Of course, we have omitted the main conclusions of these studies. Following your suggestion, we have added the conclusion that elderly blood donors contribute more to blood supply and that blood safety is ensured. The following conclusions are drawn from these studies: On one hand, elderly blood donors have a stronger willingness to donate blood and their contribution to blood supply is increasing; on the other hand, no adverse effects on the health of elderly blood donors have been found due to blood donation behavior. The citation format of the research has been revised in the corresponding content in the article.

(2) Methods

Line 89-93: Should be in a separate section covering all methods sections.

Response:

We placed these contents in the section of statistical analysis methods.

Line 95-98: What is donation frequency limits in China? Please include this.

Response:

We have added the requirements of Chinese regulations on the frequency of blood donation here. The Blood Donation Law of China stipulates that the interval between two donations of whole blood should be no less than six months.

Line 113: Please include which assays are used, company etc..

Response:

We have added the description of statistical methods and included this part.

(3) Results

Table 1 should be cohort descriptives and the current table 1 would be easier to comprehend as a figure.

Response:

It has been modified as required.

Reviewer #2: Population aging has two impacts on the blood supply. On the one hand, fewer people are eligible to donate blood. On the other hand, the demand for clinical blood use has increased. In recent years, the feasibility of expanding elderly blood donors has been evaluated all over the world.

The problem is particularly prominent in China, a country that accounts for about 20% of the world's population, where the blood donation rate per thousand people was only 12.2‰ in 2024. Faced with the continued pressure on blood supply, the Chinese government relaxed the age limit for blood donors in 2012, expanding the age limit from 55 to 60 years in an effort to improve the blood availability.

However, blood is supplied to clinical patients, so its safety is of vital importance. This safety is reflected in two aspects: first, the safety of blood donors - as an elderly group, whether donating blood will bring them health risks; second, safety of blood products - whether the blood from the elderly can meet the quality requirements of transfusion medicine.

This paper is a retrospective analysis based on these two concerns. In general, the topic is meaningful, the logical structure of the writing is clear, and the conclusions are presented in an appropriate fashion and supported by the data. As an attempt of China's solution, it provides effective data support for solving China's blood supply problem, and also provides important reference for global countries.

However, the paper needs to be improved in the following aspects.

Major revision

1. Written in standard English

Many expressions in this article do not meet the requirements of standard English. It is recommended to seek a native speaker for revision. The expressions that may cause ambiguity, unclear meaning, or sentences that are too long and need to be re-sentenced include but are not limited to:

Title, L26-30, L34-35, L54-61, L64-65, L69: ecology, L70: opposing, L74-75, L88-89, L91-92, L98: increase, L102: content, L186-190, L193: promote, L201: considers, L227-L234, L239: partly, L240, L244: lipid donation, L246: hypertriglyceridemic, L244-246, L251: diet, L252: TG, L257: suitability, L281&282: strengths.

Response:

Thank you for your suggestion. After we revised the manuscript, we invited a language editor from an English-speaking country to revise the language of the entire text and make detailed modifications to the content of the specific positions you proposed. Thank you again for your patient guidance, which has been of great help in improving the quality of our manuscripts. The title of the manuscript has been revised to “Safety and Contribution of Elderly Whole Blood Donors After Raising the Upper Age Limit: Hemovigilance Data from a Chinese Region from 2012 to 2023”.

2. Terminology for transfusion medicine

Many of the terms related to transfusion medicine in this article should be standardized, such as “donation” vs donor”, “donation” vs “donation times”. The mixed use of these expressions is particularly reflected in Table 1, L208, L212-218, etc.

Another example is the inconsistency between the "TTDI" in the main text and the title of Table 5. It is recommended to unify it as TTI (transfusion-transmitted infection).

Response:

Thank you for your reminder. According to your suggestion, we have standardized the terms of the entire text. The "donations" in this article is synonymous with the "donation times" you mentioned.

3. A statistical description should be added to the Materials and Methods section.

There are many statistical inference conclusions in the results section of the article, and the methodological information of these statistical analyses should be given in the materials and methods section.

In addition, “significantly” should be used with caution, and it needs to be supported by statistical inferences, such as in the "The deferral rate of elderly donors" section.

And, does the word “lower” in L167 mean statistically lower? Such expressions should be standardized.

Response:

Thank you for your advice. In the Materials and Methods section, we have added a description of the statistical methods: All data is exported from the blood information management system after approval. At the time the authors obtained the data, donor information had been anonymized. The authors obtained information during or after data collection that could not identify individual participants. R software (version 4.2.3) was used for all statistical analyses..GraphPad software (GraphPad Prism 9) was used to produce statistical graphs. The comparison of rates was conducted using the chi-square test, and a P value less than 0.05 was considered statistically significant.

Minor revision

1. Regarding charts: It is recommended that at least Figure 2 be in color.

Response:

Thank you for your advice. We have changed Figure 2 to a color figure as per your suggestion.

2. Industry standard should be given a standard number, such as “GB 18469-2012”.

Response:

Thank you for your suggestion. We have added standard numbers to all the contents involving the names of national/industry standards in the article.

3. The age-related changes in triglyceride levels described in [23] do not appear to be sufficient to threaten the requirement for blood donation. It is necessary to carefully examine whether appropriate references have been cited. Other literatures should also be identified.

Response:

Thank you for your reminder. The relationship between plasma triglyceride levels and blood donation was indeed not mentioned in the references. The content of triglycerides can affect the appearance of plasma. Age changes can lead to changes in plasma triglyceride levels. Plasma TGs levels are higher in older adults versus younger adults. So we use the relationship between age and plasma triglyceride levels to explain the relationship between the age of blood donors and the appearance of plasma turbidity.

4. Others

L132-133: “percent” should be “%”, and L138: “II” should be “2”.

L273: “Le Chang et al” should be “Chang et al”, L275: “Ling Shiet al” should be “Shi et al”. L293: “stations” should be “banks”. L343, L370 should be the “National Health Commission of the People's Republic of China” or the “Former Ministry of Health of the People's Republic of China”.

Response:

Thank you for your suggestion. We revised the corresponding contents in the article.

---

## [Decision Letter · Decision Letter 1]

23 Jun 2025

Dear Dr. Yang,

Thank you for submitting your revised manuscript. After careful consideration, we feel that there are still some substantial parts that require revision. Therefore, we invite you to submit a revised version of the manuscript that addresses the points raised.

We look forward to receiving your revised manuscript.

Kind regards,

Rena Hirani

Academic Editor

PLOS ONE

Journal Requirements:

Reviewers' comments:

Reviewer's Responses to Questions

**Comments to the Author**

Reviewer #1: (No Response)

Reviewer #2: (No Response)

2. Is the manuscript technically sound, and do the data support the conclusions?

Reviewer #1: Yes

Reviewer #2: Yes

3. Has the statistical analysis been performed appropriately and rigorously?

Reviewer #1: Yes

Reviewer #2: Yes

4. Have the authors made all data underlying the findings in their manuscript fully available?

Reviewer #1: Yes

Reviewer #2: Yes

5. Is the manuscript presented in an intelligible fashion and written in standard English?

Reviewer #1: Yes

Reviewer #2: Yes

Reviewer #1: The authors have addressed the issues raised by the reviewers well. The study would benefit from a revision of the discussion section to highlight their findings and improve the communi-cation to the reader. In addition some details are missing on how outcomes for positive virus screening is defined. I suggest the authors define their key findings and then clearly state them in both abstract, discussion and conclusion. Currently, these get slightly lost and the manuscript would be greatly strengthened by a thorough review.

Overall, the study finds an increasing frequency of donors aged 56-60. Deferral rates are lower among older male donors compared to young, but the reverse is seen in females. In addition, VVR rates and HCV, HBV and syphilis incidences are lower in old donors, but HIV rates appear highest among older donors.

Abstract: The conclusion section in the abstract would benefit from a clearer and more pre-cise interpretation e.g donor health = VVR risk.

Main text:

Methods: Can you please clarify when and how ALT and turbidity are measures before dona-tion? Are the results available before the donation or performed on a pre-donation sample?

For the blood safety methods section can you describe how testing is performed (equipment, is NAT(?) testing performed individually or in pools)? And in the corresponding results (line 101)section briefly mention how you define positive reactivity here? Is it solely NAT test posi-tive or also isolated antibody positive?

Discussion: The first section of the discussion is very long and should either be drastically shortened or moved to the introduction. Please start the discussion by summarizing your main findings in a short and precise manor. Likewise, the conclusion section is lacking the main findings and why the authors conclude it’s safe to include elderly donors.

Deferral results in the discussion: It’s interesting and counterintuitive that older donors have higher plasma turbidity deferrals but fewer ALT deferrals when ALT can be moderately in-creased by dyslipidemia and obesity. Do donor selection criteria use the same cutoff for all donors, or an age adjusted one? I miss a discussion of this and if current selection criteria are suitable for older donors.

Specific comments:

Figures 3+5: I would suggest adding * or similar when significant difference between the groups

Line 207: Please refer to the latest blood guide (22nd edition) and the age criteria there, this is outdated and should not be used.

Line 290: Why does it impact the results? I would suggest referring to a paper on the healthy donor effect or simply state it here.

Reviewer #2: The author has revised most of the errors in accordance with the reviewers’ comments, but some issues seem to have been overlooked.

I Written in standard English

1. L66-67: “Notably, the safety of blood donors, as well as the collected blood, should be closely monitored”.

I can’t understand the meaning of this sentence, especially “collected blood”.

2. L98: “increase” limit?

3. L207-210: Why use the past tense?

By the way, it is currently in its 22nd edition. And the file name is “Guide to the preparation, use and quality assurance of blood components” instead of “Guidelines on the Preparation, Use and Quality Assurance of Blood Components”.

4. L247-248: At least, “the rate of deferred blood donation and the causes of deferred blood donation” can be: “the rate and causes of deferred blood donation”.

In addition, there are rarely expressions like “at various ages”.

5. L271: Can “poorer diet” cause “high lipid levels”?

6. L273-274: What’s TG?

Acronyms and abbreviations must be spelled out completely on initial appearance in text.

7. L303: It is recommended to: Highlights and limitations of the study.

II Others

8. L166-167: Figure 3 does not illustrate “the deferral rate was lower for older male donors than for younger donors”, or quantitative data such as 1.50%, 1.76%, etc.

In particular, the “Hb” and “Plasma turbidity” of the older donors were significantly higher than those of the younger donors.

The same goes for female (the “Hb” and “Other deferrals” of the older donors were significantly higher than those of the younger donors).

9. L273-274: Reference 23 discusses how alterations in triglyceride metabolism contribute to metabolic diseases, but does not mention the impact on plasma appearance.

10. L403: Should be: “National Health Commission of the People’s Republic of China” or “Former Ministry of Health of the People’s Republic of China”.

11. There is a lack of citation of reference [17].

All the references should be reviewed.

12. Since P has been specified in the materials and methods section, it should be uniform throughout the main text.

13. “previous blood donor” and “experienced donor” should be unified as “repeat donor”.

**Do you want your identity to be public for this peer review?** For information about this choice, including consent withdrawal, please see our Privacy Policy

Reviewer #1: No

Reviewer #2: No

---

## [Author Response · Author response to Decision Letter 2]

21 Jul 2025

Evaluation of the safety and contribution of elderly whole blood donors after raising the upper age restriction: a study of donor hemovigilance data in a region of China from 2012 to 2023

PONE-D-25-08734R1

Dear editor and reviewers:

We would like to thank the editor and all reviewers for your time and efforts in reviewing our manuscript entitled “Evaluation of the safety and contribution of elderly whole blood donors after raising the upper age restriction: a study of donor hemovigilance data in a region of China from 2012 to 2023”. Your comments and suggestions are valuable and helpful for revising and improving our paper, as well as important guidance for our studies in the future. Here, we have revised and re-submit the manuscript for your consideration. We have responded to each point brought up in your comments, and the changes are highlighted in the revised manuscript as well.

Reviewer #1: 

1.Abstract: The conclusion section in the abstract would benefit from a clearer and more pre-cise interpretation e.g donor health = VVR risk.

Response:

Thank you for your suggestion. We have added an explanation of donor health in the methods section of the abstract and rewritten the conclusion content according to your suggestions to make the description more accurate.

Main text:

Methods:

2.Can you please clarify when and how ALT and turbidity are measures before dona-tion? Are the results available before the donation or performed on a pre-donation sample?

Response:

Thank you for your reminder. After the health consultation for blood donors was completed, we collected a sample to quantitatively and rapidly test the ALT and plasma turbidity of each donor. The doctor determined whether the blood donation would be delayed based on the test results. We have added these contents to the method.

3. For the blood safety methods section can you describe how testing is performed (equipment, is NAT(?) testing performed individually or in pools)? And in the corresponding results (line 101)section briefly mention how you define positive reactivity here? Is it solely NAT test posi-tive or also isolated antibody positive?

Response:

Thank you for your nice advice. The blood samples of donors were simultaneously tested by serological enzyme-linked immunosorbent assay (ELISA) and viral nucleic acid test (NAT). NAT testing performed individually by Tigris (Grifols). Blood with reactive ELISA or NAT test results was judged as unqualified.

4. Discussion: The first section of the discussion is very long and should either be drastically shortened or moved to the introduction. Please start the discussion by summarizing your main findings in a short and precise manor. Likewise, the conclusion section is lacking the main findings and why the authors conclude it’s safe to include elderly donors.

Response:

Thank you for your suggestion. We rewrote the first part of the discussion. Some content has been streamlined and a brief description of the main results has been added. The conclusion section has also been rewritten.

5. Deferral results in the discussion: It’s interesting and counterintuitive that older donors have higher plasma turbidity deferrals but fewer ALT deferrals when ALT can be moderately in-creased by dyslipidemia and obesity. Do donor selection criteria use the same cutoff for all donors, or an age adjusted one? I miss a discussion of this and if current selection criteria are suitable for older donors.

Response:

Thank you for your detailed review and interesting questions. Yes, all blood donors are judged by the same criteria for plasma turbidity. Because the standard for blood distribution is the same and will not change due to the different ages of the blood donors.

Specific comments:

6. Figures 3+5: I would suggest adding * or similar when significant difference between the groups

Response:

Thank you for your suggestion. We made modifications to Figures 3 and 5, adding annotations with statistical differences.

7. Line 207: Please refer to the latest blood guide (22nd edition) and the age criteria there, this is outdated and should not be used.

Thank you for pointing out the error in this version of the literature. To streamline the first part of the discussion, we deleted this reference.

8.Line 290: Why does it impact the results? I would suggest referring to a paper on the healthy donor effect or simply state it here.

Response:

Thank you for pointing out the shortcomings here. We made supplementary explanations in this discussion. Experienced blood donors are familiar with the blood donation process, which can reduce their fear of donating blood and thereby decrease the occurrence of VVRs.

Reviewer #2: 

I Written in standard English

1. L66-67: “Notably, the safety of blood donors, as well as the collected blood, should be closely monitored”.I can’t understand the meaning of this sentence, especially “collected blood”.

Response:

Thank you for your suggestion. We rewrote this sentence. The safety of elderly blood donors and the blood they donate needs to be given attention.

2. L98: “increase” limit?

Response:

Thank you for your reminder. We rewrote this sentence in the revised manuscript. These findings show the contribution of elderly blood donors to blood supply following the raising of the upper age restriction of blood donors.

3. L207-210: Why use the past tense?

By the way, it is currently in its 22nd edition. And the file name is “Guide to the preparation, use and quality assurance of blood components” instead of “Guidelines on the Preparation, Use and Quality Assurance of Blood Components”.

Response:

Thank you for pointing out the error in this version of the literature. To streamline the first part of the discussion, we deleted this reference.

4. L247-248: At least, “the rate of deferred blood donation and the causes of deferred blood donation” can be: “the rate and causes of deferred blood donation”.

In addition, there are rarely expressions like “at various ages”.

Response:

Thank you for your suggestion. We made corrections in the revised manuscript.

5. L271: Can “poorer diet” cause “high lipid levels”?

Response:

Thank you for your reminder. Here it refers to poor dietary habits. We made corrections in the revised manuscript.

6. L273-274: What’s TG?

Acronyms and abbreviations must be spelled out completely on initial appearance in text.

Response:

Thank you for your suggestion. Here, TG refers to triglycerides. We marked the position where it first appeared.

7. L303: It is recommended to: Highlights and limitations of the study.

Response:

Thank you for your recommendation. We have adopted your recommended content in the revised manuscript.

II Others

8. L166-167: Figure 3 does not illustrate “the deferral rate was lower for older male donors than for younger donors”, or quantitative data such as 1.50%, 1.76%, etc.

In particular, the “Hb” and “Plasma turbidity” of the older donors were significantly higher than those of the younger donors.

The same goes for female (the “Hb” and “Other deferrals” of the older donors were significantly higher than those of the younger donors).

Response:

Thank you for your suggestion. We made modifications to Figures 3 and 5, adding annotations with statistical differences.

9. L273-274: Reference 23 discusses how alterations in triglyceride metabolism contribute to metabolic diseases, but does not mention the impact on plasma appearance.

Response:

Thank you for your suggestion. We have cited this literature here to illustrate the influence of age on triglyceride metabolism. The concentration of plasma triglycerides can affect the turbidity of plasma.

10. L403: Should be: “National Health Commission of the People’s Republic of China” or “Former Ministry of Health of the People’s Republic of China”.

Response:

Thank you for your suggestion. We made corrections in the revised manuscript.

11. There is a lack of citation of reference [17].

All the references should be reviewed.

Response:

Thank you for your careful review. We made corrections in the revised manuscript and reviewed the references throughout the text.

12. Since P has been specified in the materials and methods section, it should be uniform throughout the main text.

Response:

We reviewed the full-text statistical description.

13. “previous blood donor” and “experienced donor” should be unified as “repeat donor”.

Response:

Thank you for your suggestion. We adopted the expression you suggested in the revised manuscript.

---

## [Editor Report · Decision Letter 2]

24 Jul 2025

Dear Dr. Yang,

Thank you for submitting your manuscript to PLOS ONE. After careful consideration, we feel that it has merit but does not fully meet PLOS ONE’s publication criteria as it currently stands. Therefore, we invite you to submit a revised version of the manuscript that addresses the points raised during the review process.

Please see below for all requested changes prior to a final decision being made.

We look forward to receiving your revised manuscript.

Kind regards,

Rena Hirani

Academic Editor

PLOS ONE

Journal Requirements:

Additional Editor Comments:

Thank you for reviewing the manuscript using previous review comments.

There are still some revisions that need to be made prior to a final decision being made.

Abstract

Thid is much improved - please change line 31 to say the overall total proportions of elderly blood donations increased 10-fold. 'All blood donations' is not the correct term as you only examining donors who perform whole blood donation. All blood donations imply other types of donations such as plasma and platelets as well as whole blood.

Introduction

line 67 remove extra full stop

please consider moving lines 71-74 before line 67 'current blood transfusion.....'

Methods

line 95 - please put here the allowable donation frequency for your centre?

is the last allowable age when each donor turns 60? If so, this should be stated.

line 106 - which exact test is used to perform plasma turbidity and ALT or place any references for these tests. what was the readout values you used and please state that the same values were used for every donor regardless of age.

line 121 if NAT is unqualified how is it later confirmed to be positive. Are results from additional testing returned to the donor centre? If not, then possibly consider saying in your results about the TTID that they are not positive but unconfirmed reactives?

Results

You provide limited p value data to show significance...please put these in next to your % value for each comparison whilst in the figure it should be in the main text and also put as a value if available rather than >0.05

you never reference supplementary tables anywhere in the results. please put a reference to them where they are supposed to be viewed for readers.

line 140 - are there any significant differences in the total number of male and female donors analysed by age group? If so, these should be reported.

line 139-141 repeat of line 167-169...please check the manuscript carefully for repeated information

Please consider removing figure 1, it does not seem helpful to your message. The data is clear from the presentation in table 1.

in your supplementary table 1 you indicate the volume of each donation provided which is useful. Are there any differences of value that should be raised in the main text.

Please also reference supplementary table 1 in your manuscript.

Please check the file for supplementary table 1 to make sure only one sheet is given

Does volume donated influence the VVR rates? I suspect there is some influence.

line 182-285 is unclear what do you mean by small amount of vvr data?

line 190 - can you say positive if you say they are unqualified in the methods? might be advisable to change to reactivity rate was higher?

line 192-193 - what is the p value? doesn't seem significant? Therefore, it may not be higher?

line 197 says HIV screening reactivity if higher in younger donors but the figure shows opposite. please check

Discussion

it is long can you make some more concise statements and perhaps refrain from repeating the results once again. you should put your findings into context without repeating the % values found. for example you can clearly state that older females had higher reactivity to HIV. We theorise that this occurs due to....

line 206 - you should say 'adjusting donor age to 60 on blood donation health'

line 215 - you need to state that this was in older female donors

line 217-219 repeat of 210-212.

line 221-225 - this is due to allowable donation frequency though and the Australian rate was probably including plasma/platelet donation which increases the frequency

how have you told potential donors in age range 56-60 that they are eligible to donate? numbers may be small as they don't know they can donate?

line 239 - please specify young male donors in that line

supplementary files

please check each file and only provide the sheet that has the information you want to use

supp table 2 - which bit of data are you trying to provide

supp table 3 - please move sheet 2 and 3. please don't use abbreviations. you should put the data together as a proper table not separate lines in the excel

supp table 4 - what value does this have as you already have it as a figure 4

supp table 5 - why have you put ALT again? please also make into a proper table even if presented in excel

---

## [Author Response · Author response to Decision Letter 3]

15 Aug 2025

Evaluation of the safety and contribution of elderly whole blood donors after raising the upper age restriction: a study of donor hemovigilance data in a region of China from 2012 to 2023

PONE-D-25-08734R3

Dear editor:

Thanks very much for your kind work of our manuscript entitled “Safety and Contribution of Elderly Whole Blood Donors After Raising the Upper Age Limit: Hemovigilance Data from a Chinese Region from 2012 to 2023”. On behalf of my co-authors, we would like to express our great appreciation to editor and reviewers. At your suggestion, we have revised this manuscript and the new revised version has been completed. After the revision, the quality of the manuscript has been significantly improved. We have responded to each point brought up in your comments, and the changes are highlighted in the revised manuscript as well.

In addition, we previously expressed to you via email my request to add another author and received your consent. We would like to express our gratitude once again for your help. In this revised manuscript, we have added Xiaobing Zhu as a co-corresponding author.

1.Abstract

Thid is much improved - please change line 31 to say the overall total proportions of elderly blood donations increased 10-fold. 'All blood donations' is not the correct term as you only examining donors who perform whole blood donation. All blood donations imply other types of donations such as plasma and platelets as well as whole blood.

Response

Thank you for your suggestion. There was indeed an imprecise expression here. We have made the revision based on your advice. In the revised manuscript, 'All blood donations' has been changed to "whole blood".

2. Introduction

line 67 remove extra full stop

please consider moving lines 71-74 before line 67 'current blood transfusion.....'

Response

Thank you for your careful review of the manuscript. We have removed this unnecessary punctuation mark in the revised manuscript and reordered the content according to your requirements.

3. Methods

line 95 - please put here the allowable donation frequency for your centre?

is the last allowable age when each donor turns 60? If so, this should be stated.

line 106 - which exact test is used to perform plasma turbidity and ALT or place any references for these tests. what was the readout values you used and please state that the same values were used for every donor regardless of age.

line 121 if NAT is unqualified how is it later confirmed to be positive. Are results from additional testing returned to the donor centre? If not, then possibly consider saying in your results about the TTID that they are not positive but unconfirmed reactives?

Response

Thank you for your reminder. We have made the following revisions in the revised manuscript we submitted:

line 95 – Add “the interval between two donations should be no less than 6 months”.

Not all blood donors are allowed to extend to the age of 60. Only repeat donors can be extended to 60, while first-time donors still have the limit of 55. This information was explained in the final paragraph of our introduction.

line 106 – We utilized QL1000C Automatic Blood Donation Screening Analyzer (Hope Medical) to measure plasma turbidity and ALT levels. The assessment of plasma turbidity was conducted using turbidity analysis, which categorizes the degree of turbidity into three grades—Grade I (mild), Grade II (moderate), and Grade III (severe). Blood donation is permitted for donors with plasma turbidity classified as Grade I or Grade II. ALT levels were determined using the rate method, and donors are eligible to donate blood if their ALT level is ≤ 50 U/L.

line 121- Regarding the TTID results, we did not conduct a confirmatory test to determine if they were positive; instead, they were unconfirmed reactive results.

4.Results

(1) You provide limited p value data to show significance...please put these in next to your % value for each comparison whilst in the figure it should be in the main text and also put as a value if available rather than >0.05.

Response

Thank you for your suggestion. We reprocessed the full-text statistical expression and presented all P-values.

(2) you never reference supplementary tables anywhere in the results. please put a reference to them where they are supposed to be viewed for readers.

Response

Thank you for your reminder. We analyzed the results of supplementary table 1 in lines 160-164 and refer to supplementary table 1 here.

3� line 140 - are there any significant differences in the total number of male and female donors analysed by age group? If so, these should be reported.

Response

Thank you for your suggestions. On lines 148 to 150 of the revised manuscript. The number of blood donors aged 56-60 accounted for 0.83% of the total number of male and 0.73% of the total number of female blood donors respectively, and the difference was statistically significant (p <.000001).

4� line 139-141 repeat of line 167-169...please check the manuscript carefully for repeated information

Response

Thank you for your careful review. However, we have compared these two parts and found no duplicate content.

5� Please consider removing figure 1, it does not seem helpful to your message. The data is clear from the presentation in table 1.

Response

Thank you for your suggestion. We compared the content of Table 1 with that of Figure 1. Table 1 mainly shows the number of blood donors and the frequency of blood donation. Figure 1 mainly shows the trend changes in blood contributions made by blood donors aged 56 to 60 during the period from 2012 to 2023. They express different contents. After careful consideration, we suggest retaining Figure 1.

6� in your supplementary table 1 you indicate the volume of each donation provided which is useful. Are there any differences of value that should be raised in the main text.

Please also reference supplementary table 1 in your manuscript.

Please check the file for supplementary table 1 to make sure only one sheet is given

Response

Thank you for your suggestion. We have added the results of the data analysis in supplementary table 1 to the article (Lines 160-164). Among male blood donors, the proportion of those aged 56-60 who donated 400ml of whole blood was 83.72%, significantly higher than 70.10% for those aged 18-55, p < 0.000001. Similarly, among female blood donors, the proportion of those aged 56-60 who donated 400ml was 76.99%, significantly higher than 47.86% for those aged 18-55, p < 0.000001 (supplementary table 1).

(7)Does volume donated influence the VVR rates? I suspect there is some influence.

Response

Thank you for your question. Yes, donating whole blood volume can affect the incidence of VVR. Our data analysis shows that blood donors aged 56 to 60 (both male and female) have a higher proportion of donating 400ml of whole blood, but the incidence of VVR is significantly lower than that of blood donors aged 18 to 55. We added the elaboration of this content in the discussion. (lines 296-299)

(8) line 182-285 is unclear what do you mean by small amount of vvr data?

Response

Thank you for your careful review. There is indeed a problem of unclear expression here. We have deleted the relevant content.

(9) line 190 - can you say positive if you say they are unqualified in the methods? might be advisable to change to reactivity rate was higher?

Response

Thank you for your suggestion. We have changed "positive" in the text to "reactivity".

(10) line 192-193 - what is the p value? doesn't seem significant? Therefore, it may not be higher?

Response

We have added the description of the p-value in the text.

(11) line 197 says HIV screening reactivity if higher in younger donors but the figure shows opposite. please check

Response

Thank you for your careful review and for pointing out our mistakes. There are indeed typos in the expression here. We have made corrections in the revised manuscript.

5. Discussion

it is long can you make some more concise statements and perhaps refrain from repeating the results once again. you should put your findings into context without repeating the % values found. for example you can clearly state that older females had higher reactivity to HIV. We theorise that this occurs due to....

line 206 - you should say 'adjusting donor age to 60 on blood donation health'

line 215 - you need to state that this was in older female donors

line 217-219 repeat of 210-212.

line 221-225 - this is due to allowable donation frequency though and the Australian rate was probably including plasma/platelet donation which increases the frequency

how have you told potential donors in age range 56-60 that they are eligible to donate? numbers may be small as they don't know they can donate?

line 239 - please specify young male donors in that line

Response

Thank you for the good questions you raised. We revised each suggestion one by one. Indeed, we have not specifically informed every blood donor who is 55 years old or above that they can continue to donate blood until they are 60. It was only when these blood donors came for consultation that they learned they could continue donating blood. We added these contents in the discussion. (lines 243-246)

6. supplementary files

please check each file and only provide the sheet that has the information you want to use

supp table 2 - which bit of data are you trying to provide

supp table 3 - please move sheet 2 and 3. please don't use abbreviations. you should put the data together as a proper table not separate lines in the excel

supp table 4 - what value does this have as you already have it as a figure 4

supp table 5 - why have you put ALT again? please also make into a proper table even if presented in excel

Response

Thank you for your suggestion. Our original understanding of the supplementary table was to provide the original data for drawing. Now we know that it only provides the content not shown in the text. Therefore, we have reorganized the supplementary table. Only supplementary table 1 was retained because the contents of the other appendices have already been reflected in the pictures.

---

## [Editor Report · Decision Letter 3]

22 Aug 2025

Safety and Contribution of Elderly Whole Blood Donors After Raising the Upper Age Limit: Hemovigilance Data from a Chinese Region from 2012 to 2023

PONE-D-25-08734R3

Dear Dr. Yang,

We’re pleased to inform you that your manuscript has been judged scientifically suitable for publication and will be formally accepted for publication once it meets all outstanding technical requirements.

Kind regards,

Rena Hirani

Academic Editor

PLOS ONE
---

## [Editor Report · Acceptance letter]

PONE-D-25-08734R3

PLOS ONE

Dear Dr. Yang,

I'm pleased to inform you that your manuscript has been deemed suitable for publication in PLOS ONE. Congratulations! Your manuscript is now being handed over to our production team.

Kind regards,

on behalf of

Dr. Rena Hirani

Academic Editor

PLOS ONE